# Signal mining and analysis of adverse events of Brentuximab Vedotin base on FAERS and JADER databases

**Feilong Tan, Yanhua Li, Hongying Xia**◉*, **Wenjie Yin***

Department of Pharmacy, Yan'an Hospital Affiliated to Kunming Medical University, Kunming, China

* 398872956@qq.com (HX); 1965115@qq.com (WY)

## Abstract

### Objectives

Brentuximab Vedotin (BV) is a novel antibody-drug conjugate (ADC) approved for the treatment of classical Hodgkin's lymphoma and systemic anaplastic large cell lymphoma. However, as a relatively new therapeutic agent, the long-term safety profile and adverse event (AE) profile of BV require further investigation. This study aimed to identify significant and unexpected AEs associated with BV using data from the FDA Adverse Event Reporting System (FAERS) and the Japanese Adverse Drug Event Report (JADER) databases.

### Methods

Data on BV-related AEs were extracted from the FAERS and JADER databases. Signal detection was performed using the reporting odds ratio (ROR) and 95% confidence intervals (95% CI). Risk signals were categorized according to system organ classes (SOCs) and preferred terms (PTs) as defined by the Medical Dictionary for Regulatory Activities (MedDRA) version 26.0. In addition, the onset times of BV-related AEs were analyzed.

### Results

Between 2004 and 2023, a total of 19,279 and 2,561 AEs related to BV were recorded in the FAERS and JADER databases, respectively. At the SOC level, prominent signals in the FAERS database included blood and lymphatic system disorders, benign, malignant, and unspecified neoplasms (including cysts and polyps), as well as congenital, familial, and genetic disorders. In the JADER database, the most notable signals involved benign, malignant, and unspecified neoplasms, blood and lymphatic system disorders, and nervous system disorders. At the PT level, the top five signals in the FAERS database were peripheral motor neuropathy, peripheral sensory neuropathy, pneumocystis jirovecii pneumonia, febrile bone marrow aplasia, and polyneuropathy. Unexpected AEs included febrile bone marrow aplasia and Guillain-Barré syndrome. In the JADER database, the top five signals included peripheral motor neuropathy, peripheral sensory neuropathy, bacterial gastroenteritis, febrile neutropenia and pneumonia, with unexpected AEs such as left

**Data availability statement:** All relevant data are within the manuscript and its Supporting Information files.

**Funding:** This research was funded by the Kunming Health Science and Technology Talent Cultivation "Thousand Project" Programme (2023-SW (back-up)-84), Kunming Health Care Commission's Health Research Project (2023-13-01-017), The Scientific Research Fund Project of the Yunnan Provincial Department of Education (2024J0371), and Yunnan Provincial Association of Pharmacists in Healthcare Facilities Scientific Research Special Fund Project (2024YSXH06). The funders had no role in study design, data collection and analysis, decision to publish, or preparation of the manuscript.

**Competing interests:** No authors have competing interests

ventricular dysfunction, cardiomegaly, retinal detachment, and marasmus. The median onset time of AEs was 22 days (interquartile range [IQR] 7–81 days) in FAERS and 27 days (IQR 7–73 days) in JADER.

## Conclusion

The signal detection results from the FAERS and JADER databases highlight the importance of monitoring significant and unexpected AEs associated with BV, particularly in the early stages of treatment. These findings contribute to enhancing the post-marketing safety profile of BV and offer valuable insights for clinical risk management strategies.

## Introduction

Antibody-drug conjugates (ADCs) represent a class of targeted biopharmaceuticals that link cytotoxic agents to monoclonal antibodies via highly stable linkers, enabling selective and potent cancer cell killing. ADCs have emerged as a major focus in anticancer drug development [1]. Brentuximab Vedotin (BV, marketed as Adcetris) is an ADC specifically targeting the CD30 protein, co-developed by Takeda Pharmaceutical (Japan) and Seattle Genetics (USA). BV consists of brentuximab, a monoclonal antibody that binds to CD30, conjugated with the microtubule-disrupting agent monomethyl auristatin E (MMAE) via a protease-sensitive linker [2]. On August 19, 2011, the U.S. Food and Drug Administration (FDA) granted BV its first approval for the treatment of relapsed Hodgkin lymphoma (HL) and systemic anaplastic large cell lymphoma (sALCL) [3]. In 2018, the drug was extended with new indications for the treatment of previously untreated classical Hodgkin lymphoma (cHL) and CD30-expressing peripheral T-cell lymphomas (PTCL). Up until now, BV has gained global approval for seven indications, including as a first-line treatment for HL in combination with an AVD chemotherapy regimen, and for relapsed or refractory cHL following failure of at least two prior chemotherapy regimens [4]. Approved for marketing in Japan in January 2014, BV was introduced in China in May 2020, and was formally included in China's National Health Insurance in May 2023, significantly improving both its accessibility and reducing the economic burden on patients.

Although BV has shown promising therapeutic efficacy in clinical applications, clinical studies have identified common adverse events (AEs) associated with its use, including pneumonitis, neutropenia, febrile neutropenia, and peripheral neuropathy [5–7]. Most of these AEs are reversible and manageable; however, rare but potentially life-threatening AEs have been reported from case-studies, such as pulmonary toxicity [8], severe insulin resistance [9], severe cytokine release syndrome, hemophagocytic lymphohistiocytosis-like syndrome [10], acute pancreatitis [11], and diabetic ketoacidosis [12]. It is important to note that BV-related AEs can be triggered by any component of the drug [13]. However, late-onset and rare AEs are often difficult to detect in a timely and accurate manner due to limitations inherent to clinical trials, including small sample sizes, stringent inclusion criteria, and short follow-up durations. Thus, the long-term safety of BV and the full spectrum of its AEs warrant further investigation and confirmation.

Compared with data from laboratory and clinical trials, pharmacovigilance data are derived from real-world settings, offering a more accurate reflection of a drug's use within the general population. The FDA Adverse Event Reporting System (FAERS) is the largest database of spontaneous AE reports globally, with over 20 million reports to date. This extensive dataset provides invaluable insights into drug safety and adverse effects across a wide spectrum of

patient populations. In contrast, the Japan Adverse Drug Event Reporting (JADER) database focuses specifically on AEs reported in Japan, although it is comparatively smaller in scale. Despite its size, JADER remains crucial for analyzing adverse reactions to drugs marketed in Japan [14,15]. While differences exist between the datasets provided by FAERS and JADER, they can be viewed as complementary sources of information [16]. Thus, this study aimed to analyze and compare BV-induced AEs using data from both FAERS and JADER. The goal was to identify potential AE signals, thereby optimizing treatment plans for patients and providing a reference for the safe and rational use of BV in clinical practice.

## Methods

### Data source

BV-related AE reports were extracted from the FAERS database for the period from Q1 2004 to Q1 2023 (latest FAERS update), and from the JADER database from Q1 2004 to Q4 2023 (latest JADER update). The extracted data included patient demographics and management information (DEMO), drug information (DRUG), adverse events (REAC), patient outcomes (OUTC), reporting source (RPSR), drug initiation and discontinuation dates (THER), and indications for use or diagnosis (INDI). Reports were organized by CASEID, FDA_DT, and PRIMARYID in the DEMO table, following the FDA's recommended de-duplication method. Specifically, for reports with the same CASEID, the report with the largest FDA_DT value was retained, and for reports with the same CASEID and FDA_DT, the report with the largest PRIMARYID value was selected. To enhance the accuracy of the study, the search was restricted to reports where "Brentuximab Vedotin" (generic name) or "Adcetris" (brand name) was listed as the primary suspected drug. Similarly, in the JADER database, BV was identified using the search terms "ブレンツキシマブ ベドチン" (generic name) or "アドセトリス" (brand name), focusing on reports where BV was classified as a "suspected" drug. Adverse event coding was carried out using the preferred system organ class (SOC) and preferred term (PT) in accordance with the Medical Dictionary for Regulatory Activities (MedDRA) version 26.0. The FAERS and JADER data files are publicly available through their respective official website (https://fis.fda.gov/extensions/FPD-QDE-FAERS/FPD-QDE-FAERS.html; https://www.info.pmda.go.jp/fukusayoudb/CsvDownload).

### Data analysis

Signal detection was carried out using the reporting odds ratio (ROR) method, a disproportionality analysis technique [17]. The key metrics for the ROR method include the ROR value and its 95% two-sided confidence interval (95% CI), calculated based on a contingency table (Table 1). The formulas and criteria for this method are provided in Table 2. Significant AEs were defined as signals listed in the Important Medical Event (IME) list (version 27.0) published by the European Union [18], while unexpected AEs were those not included on any

**Table 1. Four-grid table for analysis.**

| Drugs | Target AEs | Non-target AEs | Total |
|---|---|---|---|
| Brentuximab Vedotin | a | b | a+b |
| Non- Brentuximab Vedotin | c | d | c+d |
| Total | a+c | b+d | N = a+b+c+d |

a, number of reports of the target AE associated with BV; b, number of reports of other AEs for BV; c, number of reports of the target AE for other drugs; d, number of reports of other AEs for other drugs.

**Table 2. Summary of major algorithms used for signal detection.**

| Algorithms | Equation | Criteria |
|---|---|---|
| ROR | $\text{ROR} = \dfrac{(a/c)}{(b/d)} = \dfrac{ad}{bc}$ <br> $95\ \text{CI} = e^{\ln(\text{ROR}) \pm 1.96\sqrt{(\frac{1}{a}+\frac{1}{b}+\frac{1}{c}+\frac{1}{d})}}$ | ROR025>1, N≥3 |

ROR, reporting odds ratio; CI, confidence interval; ROR025, the lower limit of the 95% two-sided CI of the ROR.

product labels. The time to onset of AEs was defined as the interval between the date of AE occurrence and the date of BV initiation. Reports with input errors or inaccurate dates were excluded. All data extraction, de-duplication, cleaning, and graphing were conducted using R 4.2.2 software, while the mapping, intensity calculations, and sorting of the data were performed using Microsoft Excel 2019 software.

## Results

### Basic information on AE reports

In the FAERS database, BV-related AEs were reported in 5,849 patients, accounting for 19,279 individual occurrences, while in the JADER database, 1,878 patients were associated with 2,561 BV-related AEs (Table 3). Regarding gender distribution, both databases indicated a higher prevalence of cases among males compared to females. In terms of age, the FAERS database had significant missing age data (2860, 48.90%). Excluding missing data, the FAERS database was predominantly composed of individuals aged 18–45 (1,159, 19.82%), whereas the JADER database was mainly represented by individuals aged ≥65 (811, 43.18%). The majority of reports in both databases were submitted by physicians. Geographically, most reports in the FAERS database originated from the United States, followed by Japan, while all reports in the JADER database were from Japan. The annual distribution of reports showed a rising trend followed by a decline in both databases, with the highest number of reports occurring in 2020. In both databases, the most frequently reported indication was Hodgkin's disease, accounting for 57.74% and 35.47% of cases, respectively. Concerning serious clinical outcomes, the FAERS database had the highest percentage of hospitalization cases (2,477, 42.35%), followed by death (1,368, 23.39%). In contrast, the JADER database only provided clinical results for death cases (429, 22.84%), with no information available for other outcomes.

### Signals of system organ class

In the FAERS database, AEs spanned 27 SOCs, with a particularly high frequency of general disorders and administration site conditions (n = 2,788), infections and infestations (n = 1,973), investigations (n = 1,828), gastrointestinal disorders (n = 1,670), and blood and lymphatic system disorders (n = 1,498). Twelve noteworthy SOCs were identified as meeting the signal detection threshold using the ROR method, with the most prominent signals observed in blood and lymphatic system disorders (ROR025 = 5.44), neoplasms benign, malignant, and unspecified (including cysts and polyps) (ROR025 = 4.69), and congenital, familial, and genetic disorders (ROR025 = 2.46) (Fig 1A).

In the JADER database, AEs were distributed across 20 SOCs, with a relatively high frequency in blood and lymphatic system disorders (n = 511), nervous system disorders (n = 481), investigations (n = 306), neoplasms benign, malignant, and unspecified (including cysts and polyps) (n = 279), and general disorders and administration site conditions (n = 241). Five noteworthy SOCs met the ROR criteria, including neoplasms benign, malignant,

**Table 3. Characteristics of BV from FAERS and JADER databases.**

| Characteristics, number (%) | FAERS | JADER |
|---|---|---|
| Number of events | 5849 | 1878 |
| **Gender** | | |
| Male | 2456 (41.99) | 1082 (57.61) |
| Female | 1810 (30.95) | 550 (29.29) |
| Unknown | 1583 (27.06) | 246 (13.10) |
| **Age (years)** | | |
| <18 | 149 (2.55) | 26 (1.38) |
| 18≥, <45 | 1159 (19.82) | 323 (17.20) |
| 45≥, <65 | 754 (12.89) | 587 (31.26) |
| 65≥ | 927 (15.85) | 811 (43.18) |
| Other or Unknown | 2860 (48.90) | 131 (6.98) |
| **Reporter** [a] | | |
| Physician | 3069 (52.47) | 1633 (86.95) |
| Pharmacist | 741 (12.67) | 291 (19.50) |
| Consumer | 1164 (19.90) | 14 (0.75) |
| Other health-professional | 755 (12.91) | 108 (5.75) |
| Unknown | 119 (2.03) | NA |
| Lawyer | 1 (0.02) | NA |
| **Reported Countries** (Top 3) | | |
| United States | 1705 (29.15) | NA |
| Japan | 824 (14.09) | 1878 (100.00) |
| France | 425 (7.27) | NA |
| Report year | | |
| 2011 Q3-Q4 | 33 (0.56) | NA |
| 2012 | 165 (2.82) | NA |
| 2013 | 279 (4.77) | NA |
| 2014 | 362 (6.19) | 64 (3.41) |
| 2015 | 308 (5.27) | 75 (3.99) |
| 2016 | 458 (7.83) | 75 (3.99) |
| 2017 | 484 (8.27) | 86 (4.58) |
| 2018 | 566 (9.68) | 152 (8.09) |
| 2019 | 818 (13.99) | 295 (15.71) |
| 2020 | 835 (14.28) | 406 (21.62) |
| 2021 | 827 (14.14) | 313 (16.67) |
| 2022 | 572 (9.78) | 280 (14.91) |
| 2023[b] | 142 (2.43) | 132 (7.03) |
| **Indications** (Top 5) | | |
| Hodgkin's disease | 3377 (57.74) | 666 (35.47) |
| Anaplastic large-cell lymphoma | 422 (7.21) | 114 (6.08) |
| Cutaneous T-cell lymphoma | 264 (4.52) | 20 (1.05) |
| Peripheral T-cell lymphoma | 211 (3.60) | 76 (4.04) |
| Adult T-cell lymphoma/leukaemia | 976 (16.69) | 31 (1.64) |
| **Serious outcomes** | | |
| Death | 1368 (23.39) | 429 (22.84) |
| Disability | 140 (2.39) | NA |
| Hospitalization-initial or prolonged | 2477 (42.35) | NA |
| Life-threatening | 349 (5.97) | NA |

FAERS, FDA adverse event reporting system; JADER, Japanese adverse drug event report; Q, quarter; NA, Not Applicable (for relevant criterias); a, some cases were reported by multiple reporters; b, FAERS database statistics to Q1 2023.

**A (FAERS)**

| SOC | N | ROR (95%CI) |
|---|---|---|
| General disorders and administration site conditions | 2788 | 0.90 (0.86-0.93) |
| Infections and infestations | 1973 | 2.36 (2.25-2.47) |
| Investigations | 1828 | 2.25 (2.14-2.36) |
| Gastrointestinal disorders | 1670 | 1.15 (1.09-1.20) |
| Blood and lymphatic system disorders | 1498 | 5.73 (5.44-6.05) |
| Nervous system disorders | 1476 | 1.07 (1.02-1.13) |
| Respiratory, thoracic and mediastinal disorders | 1282 | 1.61 (1.52-1.70) |
| Neoplasms benign, malignant and unspecified (incl cysts and polyps) | 1218 | 4.98 (4.69-5.27) |
| Injury, poisoning and procedural complications | 1076 | 0.69 (0.65-0.74) |
| Skin and subcutaneous tissue disorders | 814 | 0.85 (0.79-0.91) |
| Metabolism and nutrition disorders | 732 | 1.98 (1.84-2.14) |
| Cardiac disorders | 471 | 1.15 (1.05-1.26) |
| Musculoskeletal and connective tissue disorders | 397 | 0.49 (0.45-0.54) |
| Vascular disorders | 389 | 1.10 (0.99-1.22) |
| Hepatobiliary disorders | 381 | 2.56 (2.31-2.83) |
| Immune system disorders | 332 | 1.71 (1.53-1.90) |
| Renal and urinary disorders | 315 | 0.97 (0.87-1.08) |
| Psychiatric disorders | 233 | 0.30 (0.27-0.35) |
| Surgical and medical procedures | 117 | 1.00 (0.83-1.19) |
| Eye disorders | 97 | 0.37 (0.30-0.45) |
| Social circumstances | 39 | 0.66 (0.48-0.90) |
| Ear and labyrinth disorders | 37 | 0.55 (0.40-0.76) |
| Reproductive system and breast disorders | 31 | 0.45 (0.32-0.64) |
| Product issues | 25 | 0.27 (0.18-0.39) |
| Pregnancy, puerperium and perinatal conditions | 22 | 0.73 (0.48-1.10) |
| Endocrine disorders | 20 | 0.76 (0.49-1.18) |
| Congenital, familial and genetic disorders | 18 | 3.91 (2.46-6.21) |

0  1  2  3  4  5  6  7

Signal NO ← → Signal YES

**B (JADER)**

| SOC | N | ROR (95%CI) |
|---|---|---|
| Blood and lymphatic system disorders | 511 | 2.76 (2.50-3.04) |
| Nervous system disorders | 481 | 2.39 (2.16-2.64) |
| Investigations | 306 | 1.18 (1.04-1.32) |
| Neoplasms benign, malignant and unspecified (incl cysts and polyps) | 279 | 3.06 (2.70-3.47) |
| General disorders and administration site conditions | 241 | 1.86 (1.63-2.12) |
| Infections and infestations | 228 | 0.99 (0.87-1.14) |
| Respiratory, thoracic and mediastinal disorders | 113 | 0.56 (0.47-0.68) |
| Gastrointestinal disorders | 97 | 0.49 (0.40-0.60) |
| Metabolism and nutrition disorders | 58 | 0.53 (0.41-0.69) |
| Skin and subcutaneous tissue disorders | 52 | 0.31 (0.23-0.40) |
| Cardiac disorders | 50 | 0.50 (0.37-0.66) |
| Hepatobiliary disorders | 37 | 0.31 (0.23-0.43) |
| Injury, poisoning and procedural complications | 33 | 0.49 (0.35-0.69) |
| Renal and urinary disorders | 27 | 0.28 (0.19-0.40) |
| Immune system disorders | 24 | 0.30 (0.20-0.45) |
| Vascular disorders | 9 | 0.15 (0.08-0.28) |
| Musculoskeletal and connective tissue disorders | 7 | 0.11 (0.05-0.23) |
| Eye disorders | 5 | 0.14 (0.06-0.34) |
| Endocrine disorders | 2 | 0.06 (0.01-0.23) |
| Psychiatric disorders | 1 | 0.02 (0.003-0.13) |

0  1  2  3  4

Signal NO ← → Signal YES

**Fig 1. System organ distribution of AEs report numbers for BV in FAERS (A) and JADER (B) databases.**

and unspecified (including cysts and polyps) (ROR025 = 2.70), blood and lymphatic system disorders (ROR025 = 2.50), nervous system disorders (ROR025 = 2.16), general disorders and administration site conditions (ROR025 = 1.63), and investigations (ROR025 = 1.04) (Fig 1B).

## Signals of preferred terms

In this study, we applied the ROR method to mine the FAERS and JADER databases for AE signals, excluding signals unrelated to adverse drug reactions, such as injuries, poisoning, procedural complications, surgical and medical procedures, social circumstances, and product issues. A total of 373 and 29 positive risk signals were identified in the FAERS and JADER databases, respectively.

In the FAERS database, the most frequently reported AEs included death, pyrexia, peripheral neuropathy, febrile neutropenia, neutropenia, diarrhea, and pneumonia. In contrast, in the JADER database, the most commonly reported AEs were peripheral neuropathy, neutrophil count decreased, neutropenia, and febrile neutropenia. Across both databases, AEs such as peripheral neuropathy, neutropenia, febrile neutropenia, pneumonia, and sepsis were consistently observed with higher incidence and were classified as significant AEs (Figs 2A and 2B).

A comparative analysis of the top 20 PTs, ranked by ROR025 in both databases, revealed that the associated SOCs primarily involved nervous system disorders, infections and infestations, blood and lymphatic system disorders, respiratory, thoracic and mediastinal disorders, gastrointestinal disorders, and metabolism and nutrition disorders. Additionally, the JADER database included SOCs related to cardiac and eye disorders (Fig 3).

In the FAERS database, 15 of the top 20 PTs with the highest signal intensity were significant AEs, and 2 were classified as unexpected AEs. The top 5 PTs were peripheral motor neuropathy, peripheral sensory neuropathy, pneumocystis jirovecii pneumonia, febrile bone marrow aplasia, and polyneuropathy. The unexpected AEs identified were febrile bone marrow aplasia and Guillain-Barré syndrome (Fig 3A).

In the JADER database, 11 of the top 20 PTs with the highest signal intensity were significant AEs, while 4 were unexpected AEs. The top 5 PTs were peripheral motor neuropathy, peripheral neuropathy, bacterial gastroenteritis, peripheral sensory neuropathy, and febrile neutropenia. The unexpected AEs included left ventricular dysfunction, cardiomegaly, retinal detachment, and marasmus (Fig 3B).

## Onset time of BV-related AEs

The onset times of BV-associated AEs were collected from the FAERS and JADER databases (Fig 4). After excluding cases with unreported or unknown onset times, data on AE onset were available for 2,553 events (43.65%, 2,553/5,849) in the FAERS database and 741 events (39.46%, 741/1,878) in the JADER database. The median onset time of AEs in the FAERS and JADER databases was 22 days (interquartile range [IQR] 7–81 days) and 27 days (IQR 7–73 days), respectively. These findings indicate considerable variability in AE onset times, with the majority occurring within the first three months following the initiation of BV treatment. Importantly, AEs were still observed more than a year after treatment initiation, accounting for 4.15% and 2.56% of cases in the FAERS and JADER databases, respectively.

## Discussion

### Analysis of the composition of AE reports

To our knowledge, this study represents the first effort to mine and analysis BV-related data from both the FAERS and JADER databases. An analysis of basic demographic information from the AE reports in these databases revealed a higher proportion of male cases compared to female cases, consistent with the global male-to-female ratio of 1.44:1 for new HL cases in 2020 [19]. The

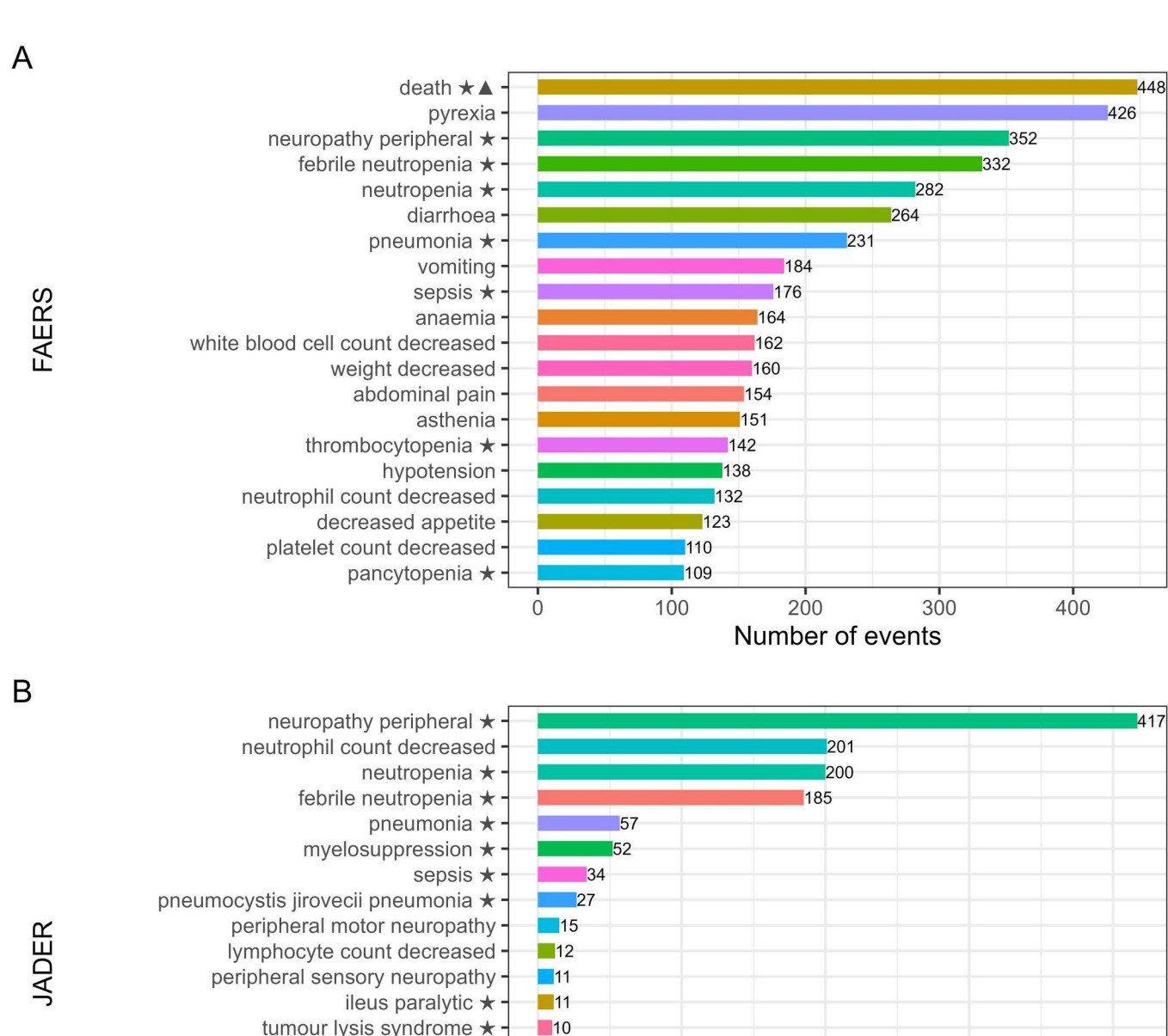

**Fig 2. The top 20 PTs ranked by the number in FAERS (A) and JADER (B) databases.** ★ indicates AE signals listed in the Important Medical Event list; ▲ indicates AE signals not documented in the label, same as below figure.The rest of the diagrams were checked and confirmed to be correct..

## A (FAERS)

| PT | SOC | a | ROR (95%CI) |
|---|---|---|---|
| peripheral motor neuropathy | Nervous system disorders | 36 | 92.32 (66.10-128.93) |
| peripheral sensory neuropathy | Nervous system disorders | 70 | 41.24 (32.54-52.27) |
| pneumocystis jirovecii pneumonia ★ | Infections and infestations | 88 | 25.34 (20.53-31.28) |
| febrile bone marrow aplasia ★▲ | Blood and lymphatic system disorders | 33 | 25.82 (18.31-36.41) |
| polyneuropathy ★ | Nervous system disorders | 77 | 22.31 (17.81-27.94) |
| febrile neutropenia ★ | Blood and lymphatic system disorders | 332 | 17.49 (15.69-19.51) |
| Guillain-Barre syndrome ★▲ | Nervous system disorders | 27 | 20.49 (14.02-29.94) |
| pulmonary toxicity ★ | Respiratory, thoracic and mediastinal disorders | 36 | 19.34 (13.92-26.85) |
| ileus paralytic ★ | Gastrointestinal disorders | 23 | 19.63 (13.02-29.60) |
| neuropathy peripheral ★ | Nervous system disorders | 352 | 12.13 (10.91-13.48) |
| progressive multifocal leukoencephalopathy ★ | Infections and infestations | 35 | 13.19 (9.45-18.39) |
| neutrophil count decreased | Investigations | 132 | 11.12 (9.37-13.20) |
| myelosuppression ★ | Blood and lymphatic system disorders | 63 | 11.62 (9.07-14.89) |
| cytomegalovirus infection | Infections and infestations | 57 | 11.43 (8.81-14.83) |
| neutropenic sepsis ★ | Infections and infestations | 28 | 12.48 (8.61-18.11) |
| tumour lysis syndrome ★ | Metabolism and nutrition disorders | 30 | 11.77 (8.22-16.85) |
| lymphopenia | Blood and lymphatic system disorders | 39 | 9.09 (6.64-12.46) |
| ileus ★ | Gastrointestinal disorders | 31 | 9.11 (6.40-12.97) |
| neutropenia ★ | Blood and lymphatic system disorders | 282 | 7.07(6.29-7.96) |
| pneumonitis ★ | Respiratory, thoracic and mediastinal disorders | 60 | 7.62 (5.91-9.82) |

1 10 30 50 100 140

← Signal NO    Signal YES →

## B (JADER)

| PT | SOC | a | ROR (95%CI) |
|---|---|---|---|
| peripheral motor neuropathy | Nervous system disorders | 15 | 148.13 (85.64-256.23) |
| neuropathy peripheral ★ | Nervous system disorders | 417 | 40.22 (36.15-44.74) |
| gastroenteritis bacterial | Infections and infestations | 4 | 40.69 (14.92-110.94) |
| peripheral sensory neuropathy | Nervous system disorders | 11 | 13.33 (7.34-24.21) |
| febrile neutropenia ★ | Blood and lymphatic system disorders | 185 | 8.43 (7.26-9.80) |
| full blood count decreased | Investigations | 4 | 14.57 (5.42-39.15) |
| left ventricular dysfunction ▲ | Cardiac disorders | 4 | 13.94 (5.19-37.45) |
| neutropenia ★ | Blood and lymphatic system disorders | 200 | 5.39 (4.66-6.23) |
| neutrophil count decreased | Investigations | 201 | 4.08 (3.53-4.71) |
| cardiomegaly ▲ | Cardiac disorders | 3 | 9.34 (2.99-29.16) |
| myelosuppression ★ | Blood and lymphatic system disorders | 52 | 3.86 (2.93-5.09) |
| ileus paralytic ★ | Gastrointestinal disorders | 11 | 5.17 (2.86-9.37) |
| retinal detachment ★▲ | Eye disorders | 4 | 7.28 (2.72-19.5) |
| sepsis ★ | Infections and infestations | 34 | 3.71 (2.64-5.20) |
| marasmus ▲ | Metabolism and nutrition disorders | 3 | 7.99 (2.56-24.93) |
| tumour lysis syndrome ★ | Metabolism and nutrition disorders | 10 | 3.54 (1.90-6.60) |
| progressive multifocal leukoencephalopathy ★ | Infections and infestations | 6 | 4.15 (1.86-9.27) |
| pneumocystis jirovecii pneumonia ★ | Infections and infestations | 27 | 2.64 (1.81-3.86) |
| clostridium difficile colitis ★ | Infections and infestations | 4 | 4.29 (1.60-11.46) |
| lymphocyte count decreased | Investigations | 12 | 2.82 (1.60-4.98) |

1 20 50 100 150 200 260

← Signal NO    Signal YES →

**Fig 3. The top 20 PTs ranked by the ROR025 in FAERS (A) and JADER (B).**

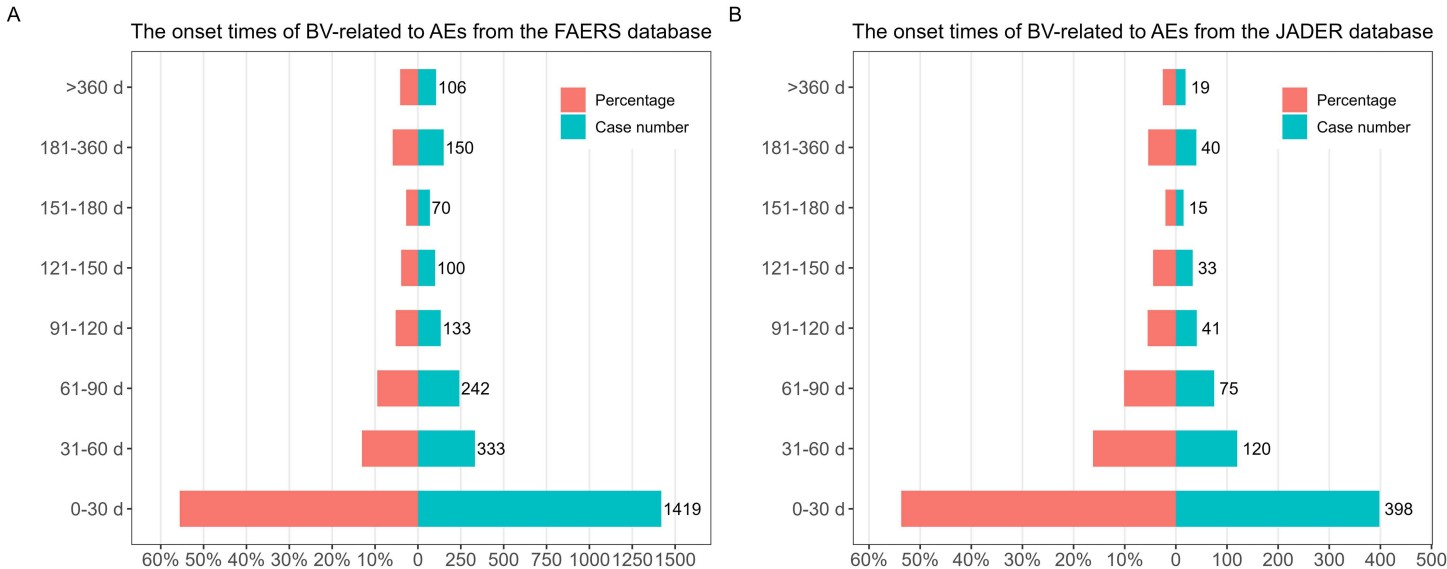

**Fig 4. The onset times of BV-related to AEs from FAERS (A) and JADER (B) databases.**

FAERS and JADER databases indicate a predominant age concentration of 18–45 years and >65 years, respectively, which is consistent with a bimodal distribution of HL, with a higher incidence in younger people and in patients aged 55 years and older [20]. In light of the potential impact of missing data on the findings of this study, further research is required to ascertain whether AEs are age-related. The majority of AE reports were submitted by healthcare professionals, indicating a degree of reliability in the data. Our study also observed a high proportion of serious clinical outcomes, such as hospitalizations and deaths, underscoring the likelihood that severe AEs are more rigorously reported and taken seriously in clinical practice.

## Risk signal mining results

In this study, signal detection was conducted using the ROR method on both the FAERS and JADER databases. The most frequently reported and noteworthy SOCs included blood and lymphatic system disorders, nervous system disorders, neoplasms benign, malignant, and unspecified (including cysts and polyps), as well as investigations. The most frequently reported PTs with the strongest signal intensities were peripheral motor neuropathy, peripheral neuropathy, peripheral sensory neuropathy, and febrile neutropenia. The signals identified were largely consistent with the known safety profile of the drug as documented in its package inserts, lending credibility to the findings of this study. It is notable that BV has minimal ROR for nervous system disorders in FAERS reported studies while it has significantly high ROR in JADER reported study. Further studies are required to ascertain whether the reasons for this discrepancy are associated with demographic factors, disease state, and dose administered. In addition to the expected AEs, this study identified several serious AEs that were not listed in the drug inserts, such as death, febrile bone marrow aplasia, Guillain-Barré syndrome, and retinal detachment, which warrant clinical attention.

## BV and neurotoxicity

Peripheral neuropathy is the most common adverse reaction to BV, which can significantly impact patients' quality of life and is a primary reason for dose adjustment or discontinuation

of therapy [21]. In a phase III clinical trial of BV for cutaneous T-cell lymphoma, 67% (44/66) of patients developed peripheral neuropathy, mostly grade 1 or 2. Moreover, 52% (23/44) of those patients required at least one dose adjustment, and 14% (9/44) permanently discontinued the drug due to neurotoxicity [22]. A higher incidence of peripheral neuropathy with ADCs containing MMAE as a payload [23], differences in neurotoxicity have been observed in ADCs containing the same MMAE. Studies have shown that the incidence of any grade of peripheral neuropathy in patients treated with BV ranged from 36% to 69%, with severe peripheral neuropathy occurring in approximately 4–14% of patients. A similar frequency was seen in patients treated with enfortumab vedotin. While polatuzumab vedotin can also cause predominantly sensory peripheral neuropathy, which is typically mild [24]. The underlying mechanism of BV-induced peripheral neurotoxicity is thought to involve the binding of MMAE to microtubule proteins, which inhibits microtubule polymerization, leading to microtubule damage and neuronal death, ultimately causing neurodegenerative lesions [25]. It is recommended that healthcare professionals closely monitor patients for symptoms of neuropathy and promptly intervene if clinical signs such as hyperalgesia, sensory hypersensitivity, burning sensations, neuropathic pain, or weakness emerge. For mild symptoms, treatment with B vitamins, gabapentin, or pregabalin may be appropriate. However, in cases of severe peripheral neurotoxicity, discontinuation or suspension of BV therapy should be considered.

## BV and haematotoxicity

Haematotoxicity is another common AE associated with BV, manifesting as febrile neutropenia, neutropenia, thrombocytopenia, pancytopenia, and myelosuppression—conditions that can be severe and significantly increase the risk of bleeding and infection. In this study, febrile neutropenia being the most frequently reported and showing the highest signal intensity. Febrile neutropenia is a serious AE that often results in severe infections, reduces the efficacy of chemotherapy, prolongs hospital stays, increases healthcare costs, and can be life-threatening in severe cases. Therefore, it is recommended to monitor complete blood counts before each administration of BV, particularly in patients with a history of severe neutropenia. For patients with pre-existing grade 3 or 4 neutropenia, BV administration should be suspended until the AE resolves to ≤ grade 2 or baseline levels before resuming treatment. If grade 3 or 4 neutropenia develops during treatment, consider delaying dosing, reducing the dose, discontinuing therapy, or administering granulocyte colony-stimulating factor (G-CSF) prophylaxis alongside subsequent BV doses [26].

## BV and progressive multifocal leukoencephalopathy

In this study, 35 reports of progressive multifocal leukoencephalopathy (PML) were identified in the FAERS database (ROR025 = 9.45) and 6 reports in the JADER database (ROR025 = 1.86), all of which demonstrated strong signals. The black box warning in the BV prescribing information highlights that JC virus infection, leading to PML, can occur in patients treated with BV and may result in death. This aligns with the increased risk of JC virus infection noted in the drug's labeling. In one study, four out of five patients treated with BV died within weeks of being diagnosed with PML [27]. The time from initial BV exposure to the onset of PML symptoms varies, with some cases occurring within three months. The underlying mechanism may involve BV depleting activated T cells expressing CD30, thereby impairing JC virus immunosurveillance in the CNS and facilitating PML development [28].

In addition to BV treatment, other potential contributing factors include prior treatments and underlying conditions that may lead to immunosuppression. It is crucial for clinicians to remain vigilant for signs of PML, particularly when new neurological symptoms arise. If PML

is suspected, BV administration should be immediately suspended, and discontinued if the diagnosis is confirmed.

## Unexpected AE signals

**BV and death.** This study identified death as the most frequently reported unexpected AE signal in FAERS database, warranting attention in clinical practice. To date, few studies have specifically focused on BV-related deaths, and neither the FDA nor the PMDA has issued a pharmacovigilance alert regarding the lethality of BV. The available data are limited to a few clinical trials and case reports. For instance, a treatment-related death due to infection was reported in a clinical trial of BV in combination with doxorubicin, vincristine, and dacarbazine (AVD) in the treatment of advanced cHL [29]. Neeman J et al. [30] described the case of an elderly male patient with lymphoma who developed neutropenia and acute liver injury after four months of BV monotherapy, ultimately succumbing to rapid disease progression.

However, in another study involving 21 patients with HL treated with BV monotherapy, three deaths were attributed to AEs, though the investigators concluded that these fatalities were unrelated to BV treatment [31]. The potential association between BV and death may be influenced by factors such as multidrug regimens and comorbidities, highlighting the need for further research to clarify this relationship.

**BV and cardiotoxicity.** The potential risk of cardiovascular complications associated with BV, a CD30-directed antibody conjugated with a microtubule inhibitor, has garnered significant attention, particularly in the context of prolonged treatment. The cardiotoxicity of BV is primarily attributed to the microtubule inhibitory effects of MMAE, which disrupts the mitotic process by interfering with microtubule polymerization, potentially impairing the normal function of cardiac cells. Clinically, cardiotoxicity may present as arrhythmias, cardiomyopathy, heart failure, or other cardiovascular symptoms, and in severe cases, can be life-threatening.

In this study, data mining of the JADER database revealed significant cardiotoxicity signals, including left ventricular dysfunction and cardiomegaly. It is important to note that cardiotoxicity is not a universal occurrence; its incidence and severity vary among individuals. Before initiating BV treatment, physicians are advised to evaluate the patient's cardiac health and implement regular cardiac monitoring throughout the course of therapy to ensure early detection and management of potential cardiotoxic effects.

**BV and ocular toxicity.** Ocular toxicity is a significant potential adverse effect associated with the use of ADCs. A real-world study reported an incidence rate as high as 89.6%, although the underlying mechanisms remain unclear. These AEs may include, but are not limited to, blurred vision, conjunctivitis, dry eye, keratopathy, and corneal epithelial changes [32]. While most ocular AEs are mild and reversible, they can hinder the course of treatment for some patients. As a result, the FDA mandates the inclusion of a black box warning regarding the risk of ocular toxicity in the drug's package insert. Liu et al. [33] reported a rare case of BV-induced uveitis, presenting with symptoms such as blurred vision, decreased visual acuity, photophobia, and eye redness. Possible mechanisms include BV targeting CD30+ T cells in uveal tissue or an immune response triggered by MMAE, the microtubule-disrupting agent in BV.

In this study, data mining of the JADER database identified significant signals of retinal detachment. Implementing a scientifically sound ocular management plan, along with dose adjustments, can help reduce the frequency and severity of ocular AEs. It is recommended that healthcare providers monitor patients' ocular health throughout BV treatment and take

preventive measures when appropriate, such as prescribing preservative-free artificial tears, eye lubricants, or topical ocular vasoconstrictor drops prior to infusion [34].

**Others.** In addition to the signals discussed earlier, this study identified several novel signals of severe AEs. Febrile bone marrow aplasia is characterized by a significant reduction in bone marrow function, leading to impaired blood cell production and fever. This condition can result in heightened susceptibility to infection and subsequent febrile episodes due to bone marrow suppression induced by BV. Guillain-Barré syndrome is a rare autoimmune disorder, typically triggered by an infection, that results in peripheral neuropathy. It is hypothesized that BV may cause immune-mediated peripheral neuropathies, akin to those seen with TNF inhibitors [35]. While the precise causal relationship between these events and BV use remains unclear, it is crucial to monitor patients for such symptoms in clinical practice.

## Time to onset of AEs

The temporal relationship between drug administration and time to onset is of paramount importance in the assessment of drug safety. A deeper comprehension of the mechanism underlying AEs is vital, as it could facilitate the identification of specific patient groups at risk and specific risk windows in the course of treatment, and contribute to the prevention or earlier diagnosis of AEs [36]. The results of this study revealed a median time to onset of BV-associated AEs of 22 and 27 days in the FAERS and JADER databases, respectively. Most cases occurred within the first one to three months after treatment initiation, a finding consistent with two literature reviews from China that analyzed BV-related adverse reactions [37,38]. These findings underscore the importance of heightened vigilance for AEs during the first month of BV treatment. Early detection of AEs is crucial, as many of these adverse drug reactions can be life-threatening.

## Limitations

This study has several limitations. (1) The study could not identify the specific population receiving the medication, making it impossible to accurately calculate the incidence of the relevant AEs; (2) The data may not be comprehensive, and the quality of reporting varied, with potential biases such as underreporting; (3) Comorbidities, drug dosage, medical history, and other factors may influence the safety assessment, but this information was missing in many AE reports, limiting the ability to control for these variables; (4) The time to onset analysis did not capture all cases, as reports with incomplete dates were excluded, especially in the JADER database; (5) The detected signals only indicate a statistical association between the drug and the target AEs, and do not necessarily establish a causal relationship; (6) No statistical tests were carried out to measure the agreement between data generated from FARES and JADER; (7) The two databases involve a large number of unexpected AEs and INDI, and the relationship is complex, so it is difficult to achieve visualization and detailed analysis; (8) The positive signals exhibited variability in terms of their type and quantity across the two databases, thereby impeding the analysis of their temporal occurrence. (9) Due to the technical limitations of our data mining technique, we could only retrieve data from the time of database construction (Q1 2004) to the most recent data at the time of mining (Q4 2023). The analysis a time period prior to the approval of BV, consequently expanding the number of background drugs and leading to changes in the calculation of ROR. However, we believe this had only a little impact on the overall trend of the conclusions. In future studies, the time of onset of positively signaled PTs and the type of AEs observed beyond one year will be further analyzed. The chi-square test will be used for evaluation to assess whether there was a significant correlation between data from the two databases [39].

## Conclusion

This is the first comprehensive and systematic study to analyze BV-related data from the FAERS and JADER databases. A number of noteworthy AE signals associated with BV have been identified, which serve to underscore the necessity of maintaining a high level of vigilance for potential adverse reactions, particularly during the initial stages of treatment. When AEs are suspected, close attention should be paid and a timely diagnosis made. If AEs are identified, the appropriate management plan should be initiated, such as adjustment of the drug regimen, postponement of treatment or dose reduction, and timely discontinuation of the drug. In the event of an untreatable AE, a multidisciplinary consultation should be arranged immediately to explore solutions. Ongoing monitoring of drug safety profiles in real-world settings is essential to ensure patient safety and effective risk management.

## Supporting information

**S1 File. Raw data (ID PONE-D-24–42713).**
(ZIP)

## Author contributions

**Conceptualization:** Feilong Tan.

**Data curation:** Feilong Tan.

**Formal analysis:** Feilong Tan.

**Funding acquisition:** Feilong Tan.

**Supervision:** hongying xia.

**Validation:** Wenjie Yin, Yanhua Li, hongying xia.

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
