## [Decision Letter · Decision Letter 0]

11 Nov 2024

PONE-D-24-42713Signal mining and analysis of adverse events of Brentuximab Vedotin base on  FAERS   and JADER databasePLOS ONE

Dear Dr. xia,

Thank you for submitting your manuscript to PLOS ONE. After careful consideration, we feel that it has merit but does not fully meet PLOS ONE’s publication criteria as it currently stands. Therefore, we invite you to submit a revised version of the manuscript that addresses the points raised during the review process. **In this paper, the authors used two electronic databases to report the adverse events associated with brentuximab vedotin**

**To begin with, brentuximab vedotin is an ADC that consists of monomethyl auristatin E (MMAE) that conjugates to CD30.**

**These adverse effects are related to MMAE; therefore, these findings may be compared to the adverse effects of other MMAE-containing ADCs or may help manage or prevent their side effects. (for example polatuzumab vedotin)**

**The discussion part is a little bit repeat of the results. It would be more appropriate if the discussion were better organized and discussed based on current data.**

**For the conclusion part, what do you suggest to help manage these adverse effects and to better report and group them? I think this topic is lacking.**

We look forward to receiving your revised manuscript.

Kind regards,

Mehmet Baysal

Academic Editor

PLOS ONE

**Journal Requirements:**

This research was funded by the Kunming Health Science and Technology Talent Cultivation "Thousand Project" Programme (2023-SW (back-up)-84), Kunming Health Care Commission's Health Research Project (2023-13-01-017), The Scientific Research Fund Project of the Yunnan Provincial Department of Education (2024J0371), and Yunnan Provincial Association of Pharmacists in Healthcare Facilities Scientific Research Special Fund Project (2024YSXH06).

3. In the online submission form, you indicated that Data cannot be shared publicly for patient privacy reasons. Researchers who meet the criteria for access to confidential data may obtain data from the corresponding author (contacted at email398872956@qq.com).

Reviewers' comments:

Reviewer's Responses to Questions

**Comments to the Author**

1. Is the manuscript technically sound, and do the data support the conclusions?

Reviewer #1: Yes

Reviewer #2: Yes

Reviewer #3: Yes

2. Has the statistical analysis been performed appropriately and rigorously? 

Reviewer #1: Yes

Reviewer #2: N/A

Reviewer #3: N/A

3. Have the authors made all data underlying the findings in their manuscript fully available?

Reviewer #1: Yes

Reviewer #2: Yes

Reviewer #3: Yes

4. Is the manuscript presented in an intelligible fashion and written in standard English?

Reviewer #1: Yes

Reviewer #2: Yes

Reviewer #3: Yes

5. Review Comments to the Author

**Reviewer #1:**  In this manuscript, the authors analysed safety data from 2 AE data bases, namely the FAERS (USA) and the JADER (Japan) in relation to an approved therapeutic agent for haematologic malignancies, the mAb-drug conjugate Brentuximab Vedotin (BV).

The authors followed a standard methodology for data mining and analysis.

The results are supported by the corresponding Tables and Figures.

In the discussion section, the findings are described in a comprehensive way, where the authors also attempt to explain the results as well as the discrepancies and casual relationship.

Despite the study limitations, as clearly described and admitted in section 4.5., the paper provides convincing and clinically useful data as to the saftey profile of BV. It is important that the information is derived from both clinical trials and real world (postmarketing) data.

**Reviewer #2:**  In the current manuscript authors studied adverse events related to Brentuximab Vedotin treatment in patients. The manuscript provides broad analysis of observed adverse events for different physiological systems reported in clinical data reported by FAERS & JADER. It is expected to observe adverse events with ADCs for Blood and Lymphatics system disorders while observed findings suggests that BV Has minimal ROR for Nervous system disorders in FEARS reported studies while it has significantly high ROR in JSDER reported study. Is this discrepancy due to demography factor or the state of disease and administered dose has some role in this.

Authors provided a method to calculate ROR while authors should mention if they have used any specific software to perform these calculations.

Authors mentioned using R and Excel for performing data analysis, I think author should elaborate which analysis was performed by excel and R. Further mention any R Package used for analysis if applicable.

**Reviewer #3: ** Tan et al. present a detailed analysis of real-life adverse events associated with Brentuximab Vedotin based on data from two large databases (FADER & JADER).The manuscript is well-written. I would like to thank the authors as I believe the findings reported in the study would open the door for future research.

In order to move forward to publishing, kindly address the following comments

Major comments

- Data Source: The rationale of selecting “period from Q1 2004 to Q1 2023” remains unclear. In other words, c and d reflect a time period different from that when the Brentuximab Vedotin was utilized which introduces bias.

- Kindly add to table 1. “Indications for use or diagnosis (INDI)” i.e., how the frequency of AEs varied in Relapsed versus previously untreated. Also, the frequency of AEs in different diagnoses.

- 3.3. Signals of preferred terms; you mention that you excluded “signals unrelated to adverse drug reactions, such as injuries, poisoning, procedural complications, surgical and medical procedures, social circumstances, and product issues” however, Figure 3B still shows “femoral neck fracture” with SOC “Injury, poisoning and procedural complications”. In the case that specific PTs were excluded and not every PT under the mentioned SOC, please make it more clear in the text.

- Duplicate reporting is mentioned as a limitation. It was mentioned earlier in the manuscript that “de-duplication method”. Kindly elaborate

- Additional analysis recommended as feasible for the following:

• No statistical tests were carried to measure the agreement between data generated from FARES and JARES. Consider conducting comparative analysis for shared AEs and for PTs with positive signals.

• For the unexpected AEs, a detailed analysis of the patient population “Indications for use or diagnosis (INDI)” would provide clinical meaningfulness.

•Onset time of BV-related AEs. Kindly elaborate on the time of onset of the PTs with positive signals. Additionally, please add details on the type of AE observed at or after 1 year.

Further comments:

Introduction

- “On August 19, 2011….. systemic anaplastic large cell lymphoma (sALCL)”. It is recommended to make it clear that the initial approval was for relapsed refractory while that for previously untreated was recently granted in 2018. Since you referenced cutaneous T-cell Lymphoma in the discussion, you might consider including it in the indications mentioned.

- In page 4 “severe “ insulin resistance [9] …..ketoacidosis [12]. Please clarify that the data was obtained from case-studies. It would further support the rationale for conducting the study.

Results and Discussion

- “In terms of age, the FAERS database was predominantly composed of individuals aged 18–45 (1,159, 19.82%), whereas the JADER database was mainly represented by individuals aged ≥65 (811, 43.18%)”. The age data appears to be missing in almost 50% of cases in the FARES database, making it hard to conclude that Brentuximab Vedotin is more predominant in 18-45 yrs age group. Kindly rephrase to make this clear or elsewise remove the comment on the FARES database.

This also applies to “In terms of age distribution, the FAERS database showed the highest proportion of patients aged 18–45 years, aligning with the well-established prevalence of HL in young adults” in the discussion section.

- "Concerning serious clinical outcomes, the FAERS database had the highest percentage of hospitalization cases (2,477, 42.35%), followed by deaths (1,368, 23.39%). In contrast, the JADER database demonstrated a definitive outcome of death (429, 22.84%)" relays that the two databases were compared. Please consider rephrasing.

- Figure 2A & 2B; x-axis kindly modify "number" to “number of events”

- For the sake of clarity, it is recommended to shuffle column 1 & 2 in figure 3A & 3B i.e., PT as the first column. Please clarify what star and triangle signify in the figures.

- Consider removing the detailed monitoring/dose modification recommendation from the discussion section. You might consider adding general recommendation in the conclusion.

- “Several significant AE signals associated with Brentuximab Vedotin were identified, including death, febrile bone marrow aplasia, Guillain-Barré syndrome, retinal detachment, and femoral neck fracture.” is repeated across the manuscript. Consider rephrasing. You might choose to remove the detailed phrase from the conclusion and keep general recommendations.

6. PLOS authors have the option to publish the peer review history of their article (what does this mean? ). If published, this will include your full peer review and any attached files.

**Do you want your identity to be public for this peer review?** For information about this choice, including consent withdrawal, please see our Privacy Policy .

Reviewer #1: No

Reviewer #2: No

Reviewer #3: No

---

## [Author Response · Author response to Decision Letter 1]

30 Dec 2024

To reviewer #1:

Comment 1: In this manuscript, the authors analysed safety data from 2 AE data bases, namely the FAERS (USA) and the JADER (Japan) in relation to an approved therapeutic agent for haematologic malignancies, the mAb-drug conjugate Brentuximab Vedotin (BV).

The authors followed a standard methodology for data mining and analysis.

The results are supported by the corresponding Tables and Figures.

In the discussion section, the findings are described in a comprehensive way, where the authors also attempt to explain the results as well as the discrepancies and casual relationship.

Despite the study limitations, as clearly described and admitted in section 4.5., the paper provides convincing and clinically useful data as to the safety profile of BV. It is important that the information is derived from both clinical trials and real world (postmarketing) data.

Response 1: Thank you for your careful review and approval.

To reviewer #2:

Comment 1: In the current manuscript authors studied adverse events related to Brentuximab Vedotin treatment in patients. The manuscript provides broad analysis of observed adverse events for different physiological systems reported in clinical data reported by FAERS & JADER. It is expected to observe adverse events with ADCs for Blood and Lymphatics system disorders while observed findings suggests that BV Has minimal ROR for Nervous system disorders in FEARS reported studies while it has significantly high ROR in JADER reported study. Is this discrepancy due to demography factor or the state of disease and administered dose has some role in this?

Response 1: Thanks for your careful review and suggestion. The results of BV monotherapy for relapsed or refractory Hodgkin lymphoma: a systematic review and meta-analysis (PMID: 34323643) indicated that the most common AEs were hematological toxicities (neutropenia: 13.3-23%, anemia: 8.8-39.0%, and thrombocytopenia: 4-4.6%), and grade ≥3 peripheral neuropathy. In this study, hematological toxicity was reflected in both SOC and PT, and the frequency and intensity were high, which was consistent with the literature reports. Neurological AEs occurred with greater intensity in the two databases, with lower intensity in the FAERS database (n = 1476, ROR = 1.07) and higher intensity in the JADER database (n = 481, ROR = 2.39) (Fig 1). Studies have shown that: Each component of the ADC, including the antibody, linker, and payload, may affect the extent of the ADC-induced toxicities (PMID: 36765668). We reviewed a large number of studies on BV-induced neurotoxicity and found that there are few reports on the analysis of factors causing differences in neurotoxicity. A study suggests that BV-induced neurotoxicity (BVIN) is a known cumulative adverse event, The level and duration of MMAE exposure in the peripheral nerve tissue is believed to be the most important determinant of PN (Peripheral Neurotoxicity). Risk factors for BVINs may include individual patient factors including demographics and patient characteristics, pre-existing conditions or prior anti-cancer therapies, concurrent medications or nutritional supplements (i.e., pyridoxine) or herbal preparations (PMID: 34885234). In view of the limited information available, we add to the discussion [It is notable that BV has minimal ROR for nervous system disorders in FAERS reported studies while it has significantly high ROR in JADER reported study. Further studies are required to ascertain whether the reasons for this discrepancy are associated with demographic factors, disease state, and dose administered.] (page 17, line 302-306).

Comment 2: Authors provided a method to calculate ROR while authors should mention if they have used any specific software to perform these calculations.

Response 2: Thanks for the suggestion. The calculations were performed by editing the ROR formula using Excel 2019. A detailed description of what analyses were performed using Excel has been provided (page 6, line 162). Function editing is shown in the figure below:

Comment 3: Authors mentioned using R and Excel for performing data analysis, I think author should elaborate which analysis was performed by excel and R. Further mention any R Package used for analysis if applicable.

Response 3: Thank you for this comment. We’ve changed [Data processing and statistical analysis were performed using Microsoft Excel 2019 and R 4.4.2.] to [All data extraction, de-duplication, cleaning, and graphing were conducted using R 4.2.2 software, while the mapping, intensity calculations, and sorting of the data were performed using Microsoft Excel 2019 software.] (page 6, line 161-164).

To reviewer #3:

Major comments:

Comment 1: Data Source: The rationale of selecting “period from Q1 2004 to Q1 2023” remains unclear. In other words, c and d reflect a time period different from that when the Brentuximab Vedotin was utilized which introduces bias.

Response 1: Thanks for your careful review and suggestion. We agree with this comment of the reviewers: the inconsistency of the time ranges extracted from the two databases does introduce bias. However, we believe that updating the FAERS database is neither feasible nor would it have a significant impact on our argument, given the costs involved. Moreover, we reviewed the literature on mining AEs using two databases and found that in some articles the time ranges extracted from the two databases were also inconsistent (e.g., PMID: 39144633, PMID: 31595301, PMID: 39635439). For this reason, we chose not to make this change, for which we apologize. We will take this into consideration when we conduct similar studies in the future, and try to adjust the time ranges as much as possible to minimize the bias, thanks again.

Comment 2: Kindly add to table 1. “Indications for use or diagnosis (INDI)” i.e., how the frequency of AEs varied in Relapsed versus previously untreated. Also, the frequency of AEs in different diagnoses.

Response 2: Thank you for this comment. The FAERS and JADER databases cover too many Indications for use or diagnosis (INDI), 161 and 712, respectively. We classified and excluded descriptions unrelated to the INDI, such as Product used for unknown indication, Not Specified, Illness, etc. The top 5 frequencies for INDI were listed and added to table 1.

Comment 3: Signals of preferred terms; you mention that you excluded “signals unrelated to adverse drug reactions, such as injuries, poisoning, procedural complications, surgical and medical procedures, social circumstances, and product issues” however, Figure 3B still shows “femoral neck fracture” with SOC “Injury, poisoning and procedural complications”. In the case that specific PTs were excluded and not every PT under the mentioned SOC, please make it more clear in the text.

Response 3: Thanks for your careful review and suggestion. We rechecked the data and found that the “Injury, poisoning and procedural complications” system was not fully excluded, resulting in “femoral neck fracture” appeared in the figure 3B. We apologize for this error. We have corrected the error and redrawn the figure 3.

Comment 4: Duplicate reporting is mentioned as a limitation. It was mentioned earlier in the manuscript that “de-duplication method”. Kindly elaborate.

Response 4: Thanks for the suggestion. The officially recommended method of de-duplication has been described in detail, specifically [Specifically, for reports with the same CASEID, the report with the largest FDA_DT value was retained, and for reports with the same CASEID and FDA_DT, the report with the largest PRIMARYID value was selected.] (page 5, line 129-132).

Comment 5: Additional analysis recommended as feasible for the following:

• No statistical tests were carried to measure the agreement between data generated from FARES and JARES. Consider conducting comparative analysis for shared AEs and for PTs with positive signals.

• For the unexpected AEs, a detailed analysis of the patient population “Indications for use or diagnosis (INDI)” would provide clinical meaningfulness.

• Onset time of BV-related AEs. Kindly elaborate on the time of onset of the PTs with positive signals. Additionally, please add details on the type of AE observed at or after 1 year.

Response 5: Thank you for the valuable suggestion. We agree with the reviewers that adding the analysis mentioned above would give the manuscript more depth. Firstly, thank you to the reviewer for prompting us to measure the consistency between the data generated by FARES and JADER through statistical tests. Our team was not previously aware of this idea, but through a literature review, we found a paper that used the chi-square test to assess whether there was a significant correlation between data from the two databases [PMID: 39402995]. We apologize that we were unable to assess the consistency of the data this time due to the difficulty of the computational operation. Secondly, the two databases involve a large number of unexpected AEs and Indications for use or diagnosis (INDI), and the relationship is complex, so it is difficult to achieve visualization and detailed analysis. Finally, the occurrence time of AEs is seriously missing in both databases, especially in the JADER database. The positive signals in the two databases are different in type and quantity, and the analysis of the occurrence time of the positive signals in the two databases has not been reported, so it is difficult to operate. In the future study, we will perform related studies for deeply and thoroughly understand those valuable suggestion. Thank you very much for this creative idea again.

Further comments:

Introduction

Comment 6: “On August 19, 2011….. systemic anaplastic large cell lymphoma (sALCL)”. It is recommended to make it clear that the initial approval was for relapsed refractory while that for previously untreated was recently granted in 2018. Since you referenced cutaneous T-cell Lymphoma in the discussion, you might consider including it in the indications mentioned.

Response 6: We are grateful for the comment. A meticulous examination of the drug's FDA marketing history and approved indications has been conducted, and the paragraph description has been revised in accordance with the necessary amendments.

We’ve changed [On August 19, 2011, the U.S. Food and Drug Administration (FDA) granted BV its first approval for the treatment of classical Hodgkin lymphoma (cHL) and systemic anaplastic large cell lymphoma (sALCL).] to [On August 19, 2011, the U.S. Food and Drug Administration (FDA) granted BV its first approval for the treatment of relapsed Hodgkin lymphoma (HL) and systemic anaplastic large cell lymphoma (sALCL). In 2018, the drug was extended with new indications for the treatment of Previously untreated classical Hodgkin lymphoma (cHL) and CD30-expressing peripheral T-cell lymphomas (PTCL).] (page 3, line 78-83).

Comment 7: In page 4 “severe “ insulin resistance [9] …..ketoacidosis [12]. Please clarify that the data was obtained from case-studies. It would further support the rationale for conducting the study.

Response 7: We are grateful for the suggestion. We have changed [Most of these AEs are reversible and manageable; however, rare but potentially life-threatening AEs have been reported, such as …..] to [Most of these AEs are reversible and manageable; however, rare but potentially life-threatening AEs have been reported from case-studies, such as…..] (page 4, line 94-98).

Results and Discussion

Comment 8: “In terms of age, the FAERS database was predominantly composed of individuals aged 18–45 (1,159, 19.82%), whereas the JADER database was mainly represented by individuals aged ≥65 (811, 43.18%)”. The age data appears to be missing in almost 50% of cases in the FARES database, making it hard to conclude that Brentuximab Vedotin is more predominant in 18-45 yrs age group. Kindly rephrase to make this clear or elsewise remove the comment on the FARES database.

This also applies to “In terms of age distribution, the FAERS database showed the highest proportion of patients aged 18–45 years, aligning with the well-established prevalence of HL in young adults” in the discussion section.

Response 8: Thanks for the reviewer careful review and suggestion. The age distribution is re-described in the results and analysed accordingly in the discussion. New references changed see [20] Ansell SM. Hodgkin lymphoma: 2025 update on diagnosis, risk-stratification, and management. Am J Hematol. 2024; 99(12):2367-2378. https://doi.org/10.1002/ajh.27470 PMID: 39239794. We have changed [In terms of age, the FAERS database was predominantly composed of individuals aged 18–45 (1,159, 19.82%), whereas the JADER database was mainly represented by individuals aged ≥65 (811, 43.18%).] to [In terms of age, the FAERS database had significant missing age data (2860, 48.90%). Excluding missing data, the FAERS database was predominantly composed of individuals aged 18–45 (1,159, 19.82%), whereas the JADER database was mainly represented by individuals aged ≥65 (811, 43.18%).] (page 7, line 180-183) in results, and we have changed [In terms of age distribution, the FAERS database showed the highest proportion of patients aged 18–45 years, aligning with the well-established prevalence of HL in young adults [20]. In contrast, the JADER database had a higher proportion of patients aged ≥65 years, a finding that warrants further epidemiological investigation.] to [The FAERS and JADER databases indicate a predominant age concentration of 18-45 years and >65 years, respectively, which is consistent with a bimodal distribution of HL, with a higher incidence in younger people and in patients aged 55 years and older [20]. In light of the potential impact of missing data on the findings of this study, further research is required to ascertain whether AEs are age-related.] (page 17, line 302-306) in discussion.

Comment 9: "Concerning serious clinical outcomes, the FAERS database had the highest percentage of hospitalization cases (2,477, 42.35%), followed by deaths (1,368, 23.39%). In contrast, the JADER database demonstrated a definitive outcome of death (429, 22.84%)" relays that the two databases were compared. Please consider rephrasing.

Response 9: We apologize for the failure to accurately describe the serious clinical outcomes in the manuscripts. The occurrence of serious clinical outcomes in two databases has been redescribed. We have changed [Concerning serious clinical outcomes, the FAERS database had the highest percentage of hospitalization cases (2,477, 42.35%), followed by deaths (1,368, 23.39%). In contrast, the JADER database demonstrated a definitive outcome of death (429, 22.84%).] to [Concerning serious clinical outcomes, the FAERS database had the highest percentage of hospitalization cases (2,477, 42.35%), followed by death (1,368, 23.39%). In contrast, the JADER database only provided clinical results for death cases (429, 22.84%), with no information available for other outcomes.] (page 7-8, line 188-193).

Comment 10: Figure 2A & 2B; x-axis kindly modify "number" to “number of events”.

Response 10: Thanks for your careful review. We have revised it.

Comment 11: For the sake of clarity, it is recommended to shuffle column 1 & 2 in figure 3A & 3B i.e., PT as the first column. Please clarify what star and triangle signify in the figures.

Response 11: Thanks for the suggestion. We have re-drawn Fig 3A & 3B as required. An explanation of the meaning of stars and triangles in the diagram has been added.

Comment 12: Consider removing the detailed monitoring/dose modification recommendation from the discussion section. You might consider adding general recommendation in the conclusion.

Response 12: Thanks for the reviewer careful review and suggestion. We removed some of the detailed monitoring from the discussion, specifically [Patients experiencing any ocular discomfort or changes in vision during treatment should seek immediate medical attention. Physicians must weigh the benefits and risks of treatment based on the patient's condition and adjust the treatment regimen as needed] (page 22, line 424-427) and [Although the number of reported cases is relatively small, the severity of these events is notabl

---

## [Decision Letter · Decision Letter 1]

22 Jan 2025

PONE-D-24-42713R1Signal mining and analysis of adverse events of Brentuximab Vedotin base on  FAERS   and JADER databasePLOS ONE

Dear Dr. xia,

Thank you for submitting your manuscript to PLOS ONE. After careful consideration, we feel that it has merit but does not fully meet PLOS ONE’s publication criteria as it currently stands. Therefore, we invite you to submit a revised version of the manuscript that addresses the points raised during the review process.

The authors should correct Indication of the reason for the time period chosen for the study and reporting de-duplication as a limitation, and also should get an input from a statistician for the ROR analysis covering time period prior drug approval.

We look forward to receiving your revised manuscript.

Kind regards,

Mehmet Baysal

Academic Editor

PLOS ONE

Journal Requirements:

Reviewers' comments:

Reviewer's Responses to Questions

**Comments to the Author**

1. If the authors have adequately addressed your comments raised in a previous round of review and you feel that this manuscript is now acceptable for publication, you may indicate that here to bypass the “Comments to the Author” section, enter your conflict of interest statement in the “Confidential to Editor” section, and submit your "Accept" recommendation.

Reviewer #1: All comments have been addressed

Reviewer #3: (No Response)

2. Is the manuscript technically sound, and do the data support the conclusions?

Reviewer #1: Yes

Reviewer #3: Partly

3. Has the statistical analysis been performed appropriately and rigorously? 

Reviewer #1: N/A

Reviewer #3: No

4. Have the authors made all data underlying the findings in their manuscript fully available?

Reviewer #1: Yes

Reviewer #3: Yes

5. Is the manuscript presented in an intelligible fashion and written in standard English?

Reviewer #1: Yes

Reviewer #3: (No Response)

6. Review Comments to the Author

Reviewer #1: Going through the authors' response to the reviewers' comments, I believe that these had been taken into the account and the manuscript is significantly improved.

Reviewer #3: I would like to thank the authors for taking several comments into consideration and updating the manuscript accordingly. However, major comments need to be addressed in order for the manuscript to be accepted.

Major comment 1 (Reviewer 3): I am herein clarifying this major comment and request that the authors kindly address it. As mentioned in the manuscript, Brentuximab Vedotin was first approved in 2011. It remains unclear why the analysis included the time period from 2004 to 2011. Your analysis is based on a relationship between AEs reported for Brentuximab Vedotin compared to that for other drugs during the same time period. It is expected that there were no AEs reported for Brentuximab Vedotin before 2011; hence bias is introduced into the equation. I believe the ROR calculation will change if events were considered only starting the introduction of Brentuximab Vedotin i.e., starting 2011.

The difference between FARES and JADER i.e., Q1 vs Q4 2023 is not a major concern

Major comment 4 (Reviewer 3) Thanks for clarifying the de-duplication method. The question remains, if you applied de-duplication, why is duplicate reporting a limitation?

Major comment 5 (Reviewer 3) Thank you for your response. It is quite understandable that further analysis might not be feasible. I appreciate that you add your response to the discussion section as a limitation and/or future considerations.

7. PLOS authors have the option to publish the peer review history of their article (what does this mean? ). If published, this will include your full peer review and any attached files.

**Do you want your identity to be public for this peer review?** For information about this choice, including consent withdrawal, please see our Privacy Policy .

Reviewer #1: No

Reviewer #3: **Yes: ** Sarah H. Youssef

---

## [Author Response · Author response to Decision Letter 2]

20 Feb 2025

Dear reviewers and editors,

Thank you for your letter and for the reviewers’ comments concerning our manuscript entitled “Signal mining and analysis of adverse events of Brentuximab Vedotin base on FAERS and JADER database” (ID: PONE-D-24-42713). Those comments are all valuable and very helpful for revising and improving our paper, as well as the important guiding significance to our researches. We have studied comments carefully and have made corrections which we hope to meet with approval. Revised portions are marked in red on the paper. The corrections in the paper and the responses to the reviewer’s comments are as flowing:

To reviewer #1:

Comment 1: Going through the authors' response to the reviewers' comments, l believe that these had been taken into the account and the manuscript is significantly improved.

Response 1: Thank you for your careful review and approval.

To reviewer #3:

Major comments:

Comment 1: I am herein clarifying this major comment and request that the authors kindly address it. As mentioned in the manuscript, Brentuximab Vedotin was first approved in 2011. lt remains unclear why the analysis included the time period from 2004 to 2011. Your analysis is based on a relationship between AEs reported for Brentuximab Vedotin compared to that for other drugs during the same time period. lt is expected that there were no AEs reported for Brentuximab Vedotin before 2011; hence bias is introduced into the equation. I believe the ROR calculation will change if events were considered only starting the introduction of Brentuximab Vedotin i.e., starting 2011.

The difference between FARES and JADER i.e., Q1 vs Q4 2023 is not a major concern.

Response 1: Thanks for clarifying the comment again. We agree that it would be more reasonable to set the starting point for data retrieval at the quarter in which the month in which the drug was launched. However, due to the technical limitations of our data mining, we were unable to change the time frame for data retrieval from the JADER database, and therefore could only retrieve data from the time of database construction (Q1 2004) to the most recent data at the time of mining (Q4 2023). In order to try to keep the excavated data as consistent as possible, we also pulled data from the FAERS database from 2004 to 2011. As questioned by the reviewers, this may have expanded the number of background drugs, which in turn led to changes in the calculation of ROR. We asked the relevant statisticians and they agreed that the calculation of the ROR would change but would have little impact on the overall trend of the conclusions. We also reviewed a large amount of related literature and found that in some articles the drugs were marketed after 2004, but the starting point of their data mining was set to Q1 2004, such as sorafenib (first marketed in 2005, PMID: 39144633), eltrombopag (first marketed in 2008, PMID: 39144633), tremelimumab (first marketed in 2022, PMID: 39144633), durvalumab (earliest marketed 2017, PMID: 39696525), and upadacitinib (earliest marketed 2019, PMID: 37663269). We will take this into account when conducting similar studies in the future, optimize our data mining techniques, and try to adjust the timeframe to minimize bias, thanks again.

Comment 4: Thanks for clarifying the de-duplication method. The question remains, if you applied de-duplication, why is duplicate reporting a limitation?

Response 4: Thank you for the valuable suggestion. We did utilize the official FDA recommended de-duplication method for de-duplication and duplicate reports are no longer a limitation. We’ve changed [The data may not be comprehensive, and the quality of reporting varied, with potential biases such as duplicate reporting and underreporting] to [The data may not be comprehensive, and the quality of reporting varied, with potential biases such as underreporting] (page 21, line 438-439).

Comment 5: Thank you for your response. It is quite understandable that further analysis might not be feasible. I appreciate that you add your response to the discussion section as a limitation and/or future considerations.

Response 5: Thank you for your understanding and satisfaction with our response to this comment. We’ve changed [The time to onset analysis did not capture all cases, as reports with incomplete dates were excluded] to [The time to onset analysis did not capture all cases, as reports with incomplete dates were excluded, especially in the JADER database] (page 21, line 442-444). We are also adding [(6) No statistical tests were carried out to measure the agreement between data generated from FARES and JADER; (7) The two databases involve a large number of unexpected AEs and INDI, and the relationship is complex, so it is difficult to achieve visualization and detailed analysis; (8) The positive signals exhibited variability in terms of their type and quantity across the two databases, thereby impeding the analysis of their temporal occurrence. In future studies, the time of onset of positively signaled PTs and the type of AEs observed beyond one year will be further analyzed. The chi-square test will be used for evaluation to assess whether there was a significant correlation between data from the two databases.] (page 21-22, line 445-454) to the discussion adding literature 39 [PMID: 39402995].

To associate Editors:

Comment 1: The authors should correct Indication of the reason for the time period chosen for the study and reporting de-duplication as a limitation and also should get an input from a statistician for the ROR analysis covering time period prior drug approval.

Response 1: Thank you for the valuable suggestion. As reviewer 3 asked the question, we agree that it would be more reasonable to set the starting point for data retrieval at the quarter in which the month in which the drug was launched. However, due to the technical limitations of our data mining, we were unable to change the time frame for data retrieval from the JADER database, and therefore could only retrieve data from the time of database construction (Q1 2004) to the most recent data at the time of mining (Q4 2023). In order to try to keep the excavated data as consistent as possible, we also pulled data from the FAERS database from 2004 to 2011. As questioned by the reviewers, this may have expanded the number of background drugs, which in turn led to changes in the calculation of ROR. We asked the relevant statisticians and they agreed that the calculation of the ROR would change but would have little impact on the overall trend of the conclusions. We also reviewed a large amount of related literature and found that in some articles the drugs were marketed after 2004, but the starting point of their data mining was set to Q1 2004, such as sorafenib (first marketed in 2005, PMID: 39144633), eltrombopag (first marketed in 2008, PMID: 39144633), tremelimumab (first marketed in 2022, PMID: 39144633), durvalumab (earliest marketed 2017, PMID: 39696525), and upadacitinib (earliest marketed 2019, PMID: 37663269). We will take this into account when conducting similar studies in the future, optimize our data mining techniques, and try to adjust the timeframe to minimize bias.

We did utilize the official FDA recommended de-duplication method for de-duplication and duplicate reports are no longer a limitation. We’ve changed [The data may not be comprehensive, and the quality of reporting varied, with potential biases such as duplicate reporting and underreporting] to [The data may not be comprehensive, and the quality of reporting varied, with potential biases such as underreporting] (page 21, line 438-439).

Thank you very much for your attention and time. Look forward to hearing from you.

Yours sincerely,

Hongying Xia

Feb 17th, 2025

Department of Pharmacy, Yan'an Hospital Affiliated to Kunming Medical University, Kunming, China.

Tel: 0871-63211154

E-mail: 398872956@qq.com

---

## [Decision Letter · Decision Letter 2]

9 Mar 2025

PONE-D-24-42713R2

Signal mining and analysis of adverse events of Brentuximab Vedotin base on  FAERS   and JADER database

PLOS ONE

Dear Dr. xia,

Thank you for submitting your manuscript to PLOS ONE. After careful consideration, we feel that it has merit but does not fully meet PLOS ONE’s publication criteria as it currently stands. Therefore, we invite you to submit a revised version of the manuscript that addresses the points raised during the review process.

All quotes have been addressed however, kindly include your justification in the section on limitations as the reviewer requested.

We look forward to receiving your revised manuscript.

Kind regards,

Mehmet Baysal

Academic Editor

PLOS ONE

Journal Requirements:

Reviewers' comments:

Reviewer's Responses to Questions

**Comments to the Author**

1. If the authors have adequately addressed your comments raised in a previous round of review and you feel that this manuscript is now acceptable for publication, you may indicate that here to bypass the “Comments to the Author” section, enter your conflict of interest statement in the “Confidential to Editor” section, and submit your "Accept" recommendation.

Reviewer #1: All comments have been addressed

Reviewer #3: All comments have been addressed

2. Is the manuscript technically sound, and do the data support the conclusions?

Reviewer #1: Yes

Reviewer #3: Yes

3. Has the statistical analysis been performed appropriately and rigorously? 

Reviewer #1: Yes

Reviewer #3: Yes

4. Have the authors made all data underlying the findings in their manuscript fully available?

Reviewer #1: Yes

Reviewer #3: Yes

5. Is the manuscript presented in an intelligible fashion and written in standard English?

Reviewer #1: Yes

Reviewer #3: Yes

6. Review Comments to the Author

Reviewer #1: (No Response)

Reviewer #3: Thank you for your response to comment #1. Please add your explanation to the limitations section.

Here's my suggestion "Due to the technical limitations of our data mining technique, we could only retrieve data from the time of database construction (Q1 2004) to the most recent data at the time of mining (Q4 2023). The analysis a time period prior to the approval of Brentuximab Vedotin, consequently expanding the number of background drugs and leading to changes in the calculation of ROR. However, we believe this had only a little impact on the overall trend of the conclusions. "

7. PLOS authors have the option to publish the peer review history of their article (what does this mean? ). If published, this will include your full peer review and any attached files.

**Do you want your identity to be public for this peer review?** For information about this choice, including consent withdrawal, please see our Privacy Policy .

Reviewer #1: No

Reviewer #3: **Yes: ** Sarah Youssef

---

## [Author Response · Author response to Decision Letter 3]

19 Mar 2025

Dear reviewers and editors,

Thank you for your letter and the editor's suggested revisions to our manuscript entitled “Signal mining and analysis of Brentuximab Vedotin adverse events based on FAERS and JADER databases” (ID: PONE-D-24-42713). We have removed the uploaded duplicate files, as well as deleted the fund information section from the body of the article and added the supporting information heading at the end of the article as requested.

Thank you very much for your attention and time. Look forward to hearing from you.

Yours sincerely,

Hongying Xia

Feb 21th, 2025

Department of Pharmacy, Yan'an Hospital Affiliated to Kunming Medical University, Kunming, China.

Tel: 0871-63211154

E-mail: 398872956@qq.com

---

## [Editor Report · Decision Letter 3]

21 Mar 2025

Signal mining and analysis of adverse events of Brentuximab Vedotin base on  FAERS   and JADER database

PONE-D-24-42713R3

Dear Dr. xia,

We’re pleased to inform you that your manuscript has been judged scientifically suitable for publication and will be formally accepted for publication once it meets all outstanding technical requirements.

Kind regards,

Mehmet Baysal

Academic Editor

PLOS ONE
---

## [Editor Report · Acceptance letter]

PONE-D-24-42713R3

PLOS ONE

Dear Dr. xia,

I'm pleased to inform you that your manuscript has been deemed suitable for publication in PLOS ONE. Congratulations! Your manuscript is now being handed over to our production team.

Kind regards,

on behalf of

Dr. Mehmet Baysal

Academic Editor

PLOS ONE